# Perceptions of healthcare finance and system quality among Nigerian healthcare workers

**Blessing Osagumwendia Josiah**[1¤a]*, **Emmanuel Chukwunwike Enebeli**[2¤a], **Brontie Albertha Duncan**[3], **Lordsfavour Uzoma Anukam**[4], **Oluwadamilare Akingbade**[1¤b], **France Ncube**[5], **Chinelo Cleopatra Josiah**[6], **Eric Kelechi Alimele**[7], **Ndidi Louis Otoboyor**[8], **Oghosa Gabriel Josiah**[9], **Jemima Ufuoma Mukoro**[10], **Blessing Chiamaka Nganwuchu**[11], **Fawole Israel Opeyemi**[12], **Timothy Wale Olaosebikan**[13], **Marios Kantaris**[14]

1 Department of Research and Statistics, Institute of Nursing Research, Oshogbo, Nigeria, 2 Leeds Teaching Hospitals NHS Trust, Leeds, United Kingdom, 3 Freelance Financial Consultant, Basseterre, Saint Kitts and Nevis, 4 IUHS School of Medicine, Basseterre, Saint Kitts and Nevis, 5 Unicaf Ltd., Harare, Zimbabwe, 6 Windsor University School of Medicine, Canyon, St Kitts and Nevis, 7 Queen Margaret university, Belfast, Northern Ireland, United Kingdom, 8 Scripps Hospital San Diego, San Diego, California, United States of America, 9 University of Sunderland, Sunderland, United Kingdom, 10 Anchor University, Lagos, Nigeria, 11 Abia State University Uturu, Uturu, Nigeria, 12 Ladoke Akintola University of Technology, Ogbomoso, Nigeria, 13 Joseph Ayo Babalola University, Ikeji-Arakeji, Osun State, Nigeria, 14 Health Services and Social Policy Research Centre, Nicosia, Cyprus

¤a Current address: Department of Nursing, Turks and Caicos Islands Community College, Grand Turk, Turks and Caicos Islands
¤b Current address: Faculty of Nursing, University of Alberta, Edmonton, Canada
* josiahblessing141@gmail.com

**Data Availability Statement:** In compliance with the applicable reporting standards, the datasets supporting the conclusions of this article are

## Abstract

Nigeria's healthcare system faces significant challenges in financing and quality, impacting the delivery of services to its growing population. This study investigates healthcare workers' perceptions of these challenges and their implications for healthcare policy and practice. A cross-sectional survey was conducted with 600 healthcare professionals from eight states across Nigeria, representing a variety of healthcare occupations. Participants completed a questionnaire that assessed their perceptions of healthcare financing, quality of care, job satisfaction, and motivation using a 5-point Likert scale, closed- and open-ended questions. Descriptive statistics, Chi-squared test, and regression analysis were used to analyze the data. The findings revealed that healthcare workers were generally not satisfied with the current state of healthcare financing and system quality in Nigeria. Poor funding, inadequate infrastructure, insufficient staffing, and limited access to essential resources were identified as major challenges. These challenges contributed to low job satisfaction, demotivation, and a desire to leave the profession. Socioeconomic factors, location State of practice, professional designation (clinical vs nonclinical), clinical designation (profession), and employment type (full-time vs part-time) were found to influence healthcare workers' perceptions ($p < 0.05$). The findings indicated a need to improve healthcare workers' satisfaction and retention, and quality of care in Nigeria, by increasing healthcare funding, transparent fund management protocols, investing in infrastructure and human resource development, and addressing regional healthcare disparities. By implementing these reforms, Nigeria can

available in the Dryad Digital Repository at https://datadryad.org/stash/share/gZJC4g6cgMgi9cJhwLZalmdEPQGrdh72L7YuOWM7pps.

**Funding:** The authors received no specific funding for this work.

**Competing interests:** The authors have declared that no competing interests exist.

enhance the quality and accessibility of healthcare services and improve the health and well-being of its citizens.

## Background

Recognizing the essential role of health in societal and economic advancement, many nations have invested significantly in creating health systems that are equitable, efficient, and of high quality to serve their populations [1–4]. Health systems differ markedly across the globe in their structure, management, financing, and their capacity to deliver necessary care effectively.

The World Health Organization (WHO) rankings serve as a benchmark for assessing and comparing national health systems, focusing on key aspects such as health infrastructure, workforce, service accessibility, administrative governance, and population health outcomes. In these rankings, Nigeria has demonstrated progress, moving from 187 out of 190 countries in 2000 to 158 out of 167 countries in 2021 [5–7]. Yet, these evaluations, while informative, do not capture the full scope of the healthcare system's performance. Despite improvements in certain performance indicators like rising average life expectancy since 1960 and a decline in the fertility rate to 5.5 [8,9], Nigeria has remained near the lower end of the spectrum, primarily due to its inability to meet the needs and expectations of its primary users (patients) and to utilize its resources efficiently by leveraging the full potential of its healthcare workforce [10,11].

The healthcare system is beset with issues such as inadequate funding, with total healthcare budgets consistently falling below the recommended 5% of GDP over the past decade [12–14], and out-of-pocket payments accounting for up to 79% of healthcare expenditure for the average Nigerian [13,15]. The system suffers from poor infrastructure, with most facilities lacking basic equipment and supplies such as stethoscopes, disposables [16,17], and essential medications like Azithromycin and Nifedipine [18]. In addition, the system is plagued by a severe workforce shortage and unequal access to healthcare services [16,17,19,20].

Feedback from healthcare workers and patients is critical for guiding health policy and planning [21,22] and is essential for understanding the quality of care and patient satisfaction. These key elements are often overlooked by standard health metrics and rankings [23,24]. This study concentrates on Nigerian healthcare workers because of observed high levels of concern among them about the healthcare system, as evidenced by the high attrition rate [16,25–27]. Insights gathered from healthcare workers can inform improvements in job satisfaction and service quality, ultimately enhancing patient outcomes [26,28,29]. By assessing healthcare workers' perceptions of healthcare quality, the study evaluates the healthcare system across the country; aiding in pinpointing priority areas for intervention to fortify service quality and restore healthcare workers' confidence in the system.

## Methods

### Research design

This study employed a quantitative cross-sectional survey design to assess healthcare workers' perceptions of quality, challenges, and best financing interventions within the Nigerian healthcare system over the preceding period.

### Population and Scope

The research focuses on healthcare workers in Nigeria, a West African nation (Fig 1) with a population exceeding 220 million as of 2024 [30].

The nine States selected for the study have a combined population of 72,025,500. Nigeria has an overall estimated healthcare worker density of 19.97 per 10,000 individuals [32].

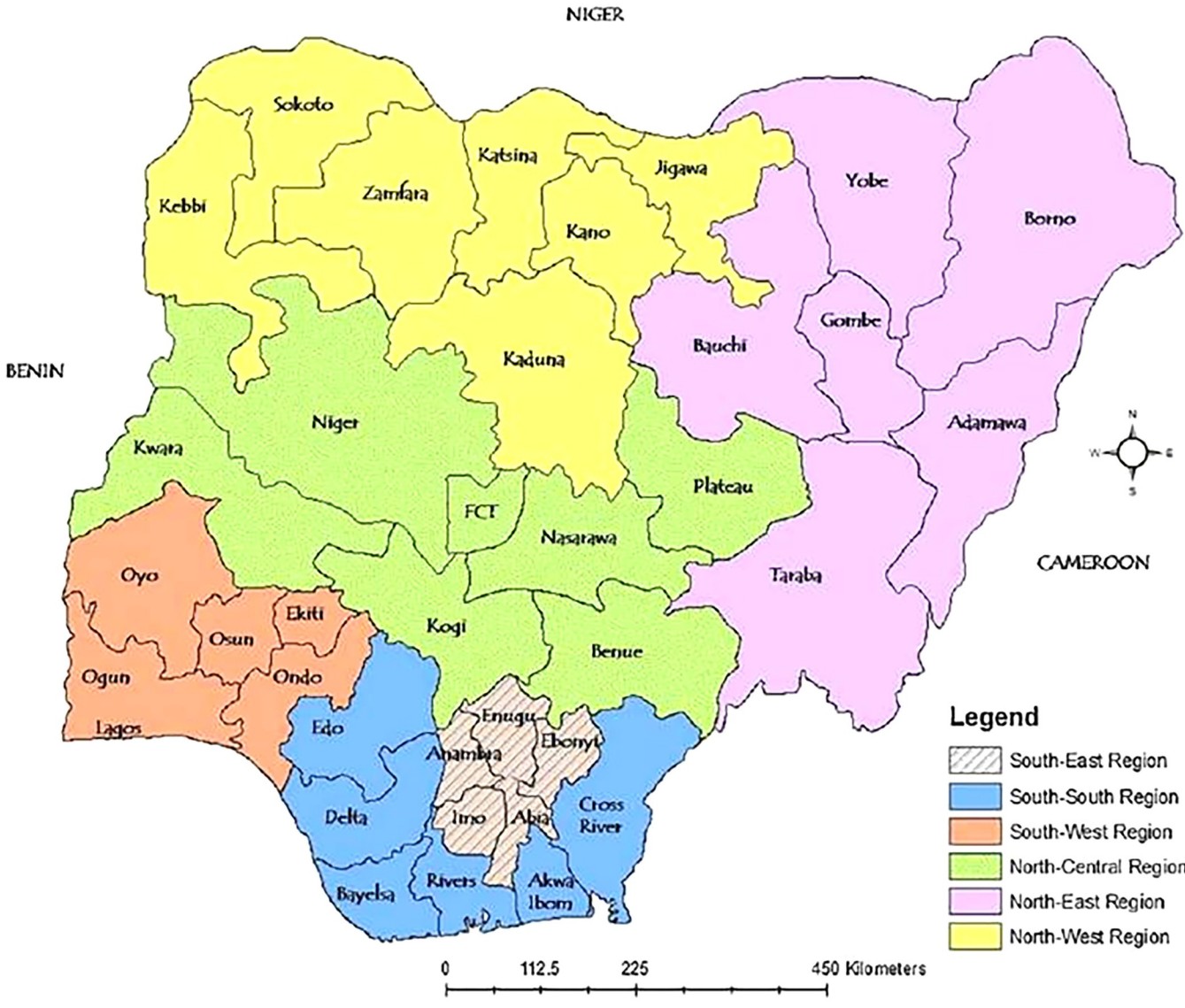

**Fig 1. Nigeria Map: Geopolitical Zones and States (*Source*: [31]).**

Therefore, the target population for the study was 143,835 healthcare workers in addition to some non-clinical healthcare workers who are working in Nigerian government health facilities currently or within the last year [32].

Healthcare workers were sampled from two randomly selected representative States in each of five geopolitical zones (except one for the South-east, the smallest geopolitical zone), capturing Nigeria's cultural and economic diversity. The North-east zone was excluded due to logistic difficulties. The States are Kwara and Plateau (North-central); Kano and Kaduna (Northwest); Lagos and Oyo (South-west); Delta and Rivers (South-south); and Enugu (South-east geopolitical zone) (Fig 1).

## Sample size determination

The sample size was calculated with the formula $Z^2(pq)/m^2$, {where Z = 1.96 at 95% Confidence Interval, p = 0.5 (incidence rate), q = 0.5 (non–incidence rate) and m = 0.05 (Margin of

**Table 1. Sample size determination.**

| State (Locations) | Total Population (N)** | ±Estimated Healthcare Workers [N/10,000)*19.97] | Targeted Sample | | |
|---|---|---|---|---|---|
| | | | Clinical | Non-Clinical | Total |
| Lagos | 13,481,800 | 26,923 | 72 | 38 | 110 |
| Delta | 5,636,000 | 11,255 | 30 | 19 | 49 |
| Kwara | 3,551,000 | 7,091 | 19 | 21 | 40 |
| Plateau | 4,717,300 | 9,420 | 25 | 12 | 37 |
| Kano | 15,462,200 | 30,878 | 82 | 34 | 116 |
| Enugu | 4,690,100 | 9,366 | 25 | 13 | 38 |
| Kaduna | 9,032,200 | 18,037 | 48 | 25 | 73 |
| Rivers | 7,478,800 | 14,935 | 40 | 31 | 71 |
| Oyo | 7,976,100 | 15,928 | 43 | 24 | 67 |
| **Total** | **72,025,500** | **143,835** | **384** | **216** | **600** |

+ Average National Healthcare workers density applied

**\*\*projected** [33].

error)} using a total healthcare workers population of 143,835 across the nine States. The resultant sample of 384, predominantly comprised of clinical healthcare workers, was proportionately selected across the nine states (Table 1). For inclusiveness, a broader range of perspectives, enhanced statistical power, and representativeness in the selected States, non-clinical healthcare workers were added to the sample.

In Nigeria, the proportion of clinical and non-clinical team members is approximately 65% to 35%, respectively, based on a pilot study conducted in Edo and Imo States. This is unlike the composition of the estimated 19.97 density reported by the WHO which significantly excludes the non-clinical staff from the measurement even though they closely interact with the healthcare system and typically make up about half of the healthcare staff [34]. An additional sample of 216 (36%) non-clinical workers was, therefore, selected during the survey to increase the initial sample size to a target of 600 participants (Table 1).

## Data collection tools

Data was collected via a structured questionnaire in print and online on Google Forms, measuring sociodemographic data, perceptions of healthcare quality, challenges, and areas of improvement as developed based on the Donabedian healthcare quality assessment model [35], the SERVIQUAL framework [36], Mosadeghrad's conceptual framework for quality of care [37] as applied by other similar studies [38], as well as the Patient Satisfaction Questionnaire (PSQ-18) which is used for the measurement of patient satisfaction with the medical care received [39]. It was composed of both yes or no questions and a Likert scale that measured the perceptions of healthcare quality. A reliability test was conducted with two samples of 25 healthcare workers from Imo and Edo States respectively with a reliability coefficient (Cronbach alpha) of 0.70.

## Data collection process

At least three government health facilities were randomly selected in each state and visited to engage the workers for data collection: one urban (capital city) and two semi-urban/rural areas. The selected settings were approached by providing the National Research Ethics Committee approval and research protocol to the hospital management teams who granted the Researchers access to their employees. Participants were randomly selected and engaged

during the data collection with print paper and online questionnaires, proof of ethical clearance, and consent forms provided to each participant to review and sign before interviews or provision of responses. Some respondents were accessed online via social media groups and emailing lists. The survey was aimed at those with about ten years of experience; hence the respondents' cut-off age was 30 years and above to ensure that they have an average of 5–10 years post-qualification.

### Data analysis method

The survey was conducted between 8/6/2023 and 20/8/2023 and the data was collated with Google sheet and transferred to Microsoft Excel and SPSS version 29 to be analyzed. The Likert scale evaluation of the Nigerian healthcare system, ratings were assigned as follows: 5 for Excellent, 4 for Good, 3 for Fair, 2 for Poor, and 1 for Very Poor. The cumulative scores of participants were then categorized into ranges: 6–10 for Very Poor, 11–15 for Poor, 16–20 for Fair, 21–25 for Good, and 26–30 for Excellent. Simple frequency distributions and percentages were applied to the sociodemographic data and various variables. Multiple linear regression analysis and Chi-square test were used to evaluate relationships between the workers' healthcare quality and system ratings and several factors like professional designations, clinical designations, type of employment, the States, and other sociodemographic attributes. The variations within each State were not examined due to the unified approach to data collection at the State level.

### Ethical consideration

The study is the healthcare workers' aspect of a compounded proposal submitted as "Critical review of healthcare financing and end-users' quality perception in Nigeria", which received approval from the Nigerian Health Research Ethics Committee (NHREC) with ID: IRB-23-018. The research is designed to separately access healthcare workers and the patient's perspectives on healthcare financing and the quality of care they receive using separate questionnaires as approved. This ethical process ensured that voluntary participation, anonymity, and secure data storage, were observed throughout this research. Also, written consents were obtained from each participant before they participated in the study.

### Inclusivity in global research

Additional information regarding the ethical, cultural, and scientific considerations specific to inclusivity in global research is included in (S1 Checklist).

## Results

This assesses healthcare workers' perceptions of healthcare financing and system quality in Nigeria over the immediate past period. A total of 638 responses were collected and 584 were validly filled. Reasons for invalidity were mainly incomplete responses. The valid responses were cleaned and analysed as shown below.

### Sociodemographic attributes

Table 2 provides a demographic breakdown of the respondents, indicating that females constitute a majority at 58.4%, while males accounted for 41.6%. The age distribution shows a significant representation in the 30–39 years bracket, comprising 39.9% of the participants. This is followed by those in the 40–49 and 50–59 years age groups, each making up 26.4%, while 60 years and above with the smallest segment had 7.4%. In terms of educational attainment, the

**Table 2. Sociodemographic data of the respondents.**

| Variables | | Frequency (N = 584) | Percentage |
|---|---|---|---|
| **Gender** | Female | 341 | 58.4 |
| | Male | 243 | 41.6 |
| **Age Group** | 30–39 years | 233 | 39.9 |
| | 40–49 years | 154 | 26.4 |
| | 50–59 years | 154 | 26.4 |
| | 60 and above | 43 | 7.4 |
| **State** | Delta | 49 | 8.4 |
| | Enugu | 35 | 6.0 |
| | Kaduna | 68 | 11.6 |
| | Kano | 93 | 15.9 |
| | Kwara | 57 | 9.8 |
| | Lagos | 104 | 17.8 |
| | Oyo | 65 | 11.1 |
| | Plateau | 31 | 5.3 |
| | Rivers | 82 | 14.0 |
| **Education** | High school or below | 53 | 9.1 |
| | Diploma | 131 | 22.4 |
| | Bachelor | 202 | 34.6 |
| | Master/Postgraduate | 130 | 22.3 |
| | PhD/Fellowships | 68 | 11.6 |
| **Profession** | Clinical staff | 445 | 76.2 |
| | Non-clinical | 139 | 23.8 |
| **Employment type** | Full-time employment | 477 | **81.7** |
| | Part-time employment | 107 | **18.3** |
| **Income (Naira)** | Less than 35,000 | 80 | 13.7 |
| | 35,000 to 49,000 | 128 | 21.9 |
| | 50,000 to 99,000 | 166 | 28.4 |
| | 100,000 to 199,000 | 147 | 25.2 |
| | 200,000 to 399,000 | 47 | 8.0 |
| | More than 400,000 | 16 | 2.7 |
| **Years of Experience** | 1–3 years | 91 | 15.6 |
| | 4–6 years | 161 | 27.6 |
| | 7–9 years | 152 | 26.0 |
| | 10 years and above | 180 | 30.8 |
| **Facility of Practice** | Government Hospital only | 349 | 59.8 |
| | Government and Private | 235 | 40.2 |

majority, 34.6%, hold a bachelor's degree, 33.9% possess some form of postgraduate degree, 22.4% have a diploma and 9.1% have attained a high school certificate or less. The professional roles of the respondents are predominantly clinical, with 76.2% working in clinical settings and 23.8% in nonclinical roles. Experience levels vary, with 15.6% having 1–3 years of experience, 27.6% with 4–6 years, 26.0% with 7–9 years, and 30.8% with 10 years or more experience.

## Rating of healthcare financing

Table 3 shows that most of the respondents gave a poor rating to all the components examined and only a few of them gave an excellent rating to the questions.

**Table 3. Respondents' Rating of healthcare financing in Nigeria (n = 584).**

| S/N | How would you rate Nigerian healthcare: | Excellent n (%) | Good n(%) | Fair n(%) | Poor n(%) | Very Poor n(%) |
|---|---|---|---|---|---|---|
| 1 | The level of health system improvements by the government | 8 (1.4) | 38 (6.5) | 141 (24.1) | 322 (55.1) | 75 (12.8) |
| 2 | Satisfaction with the healthcare management/administrative system | 8 (1.4) | 35 (6.0) | 141 (24.1) | 333 (57.0) | 67 (11.5) |
| 3 | The financing/budget for healthcare by the government | 12 (2.1) | 29 (5.0) | 134 (22.9) | 324 (55.5) | 85 (14.6) |
| 4 | New Healthcare Project Implementations in Nigeria | 11 (1.9) | 31 (5.3) | 155 (26.5) | 309 (52.9) | 78 (13.4) |
| 5 | The financial management of budgetary allocations to the healthcare sector | 8 (1.4) | 40 (6.8) | 122 (20.9) | 341 (58.4) | 73 (12.5) |
| 6 | Access to (availability of) resources needed to deliver care | 5 (0.9%) | 30 (5.1) | 131 (22.4) | 320 (55.8) | 98 (16.8) |
| | Percentage of HCW in each rating category | 1.5% | 5.8% | 23.5% | 55.6% | 13.6% |

HCW = Healthcare workers.

Table 4 shows that the ratings are heavily weighted towards the lower scores as indicated by the frequency distribution and a significant positive of 0.879. Most (73.4%) healthcare workers rated healthcare financing as *poor* and *very poor*. Only a negligible fraction (7.4%) of healthcare workers rated healthcare financing as *good* and *excellent*. A rating of *fair* was given by 19.2% of the respondents.

## Major problems in the healthcare system

**Areas of significant improvements in the healthcare.** Fig 2 highlights the major challenges in the Nigerian healthcare system as perceived by healthcare workers. The majority (90.1%) of the healthcare workers saw poor funding as a major challenge, followed closely by poor range of services (88.2%), poor staffing (87.5%), lack of proper tools and facilities (88.7%), and funds misappropriation (87.0%). Other areas reported were substandard pharmaceuticals and a lack of consumable materials in the healthcare facilities.

Fig 2 highlights key improvements in the Nigerian healthcare system, with 75.9% of respondents noting an uptick in healthcare facilities. In additional, 51.5% commend government efforts in subsidizing healthcare costs, and 51.4% acknowledge a rise in training centers for healthcare professionals.

Table 5 shows a significant majority (59.4%) of healthcare workers rated their motivation to continue working in the government healthcare facilities as either *poor* or *very poor*. And while 39% of respondents rated their willingness to encourage others to join the healthcare profession in Nigeria as average (*fair*), 52.3% have either a *poor* or *very poor* willingness to do so.

## Satisfaction with the Nigerian healthcare system

Fig 3 shows the reported satisfaction of healthcare workers with the Nigerian healthcare system. When asked if they are satisfied with the state of the healthcare system, a majority (90%) said no, while a small fraction (10%), reported yes.

**Table 4. Summary of Healthcare Workers Rating of Nigerian Healthcare Financing (n = 584).**

| Healthcare Rating | Score | Frequency | Percentage | Skewness |
|---|---|---|---|---|
| Excellent | 26–30 | 12 | 2.1 | 0.879 |
| Good | 21–25 | 31 | 5.3 | |
| Fair | 16–20 | 112 | 19.2 | |
| Poor | 11–15 | 322 | 55.1 | |
| Very Poor | 6–10 | 107 | 18.3 | |
| **Total** | | **584** | **100** | |

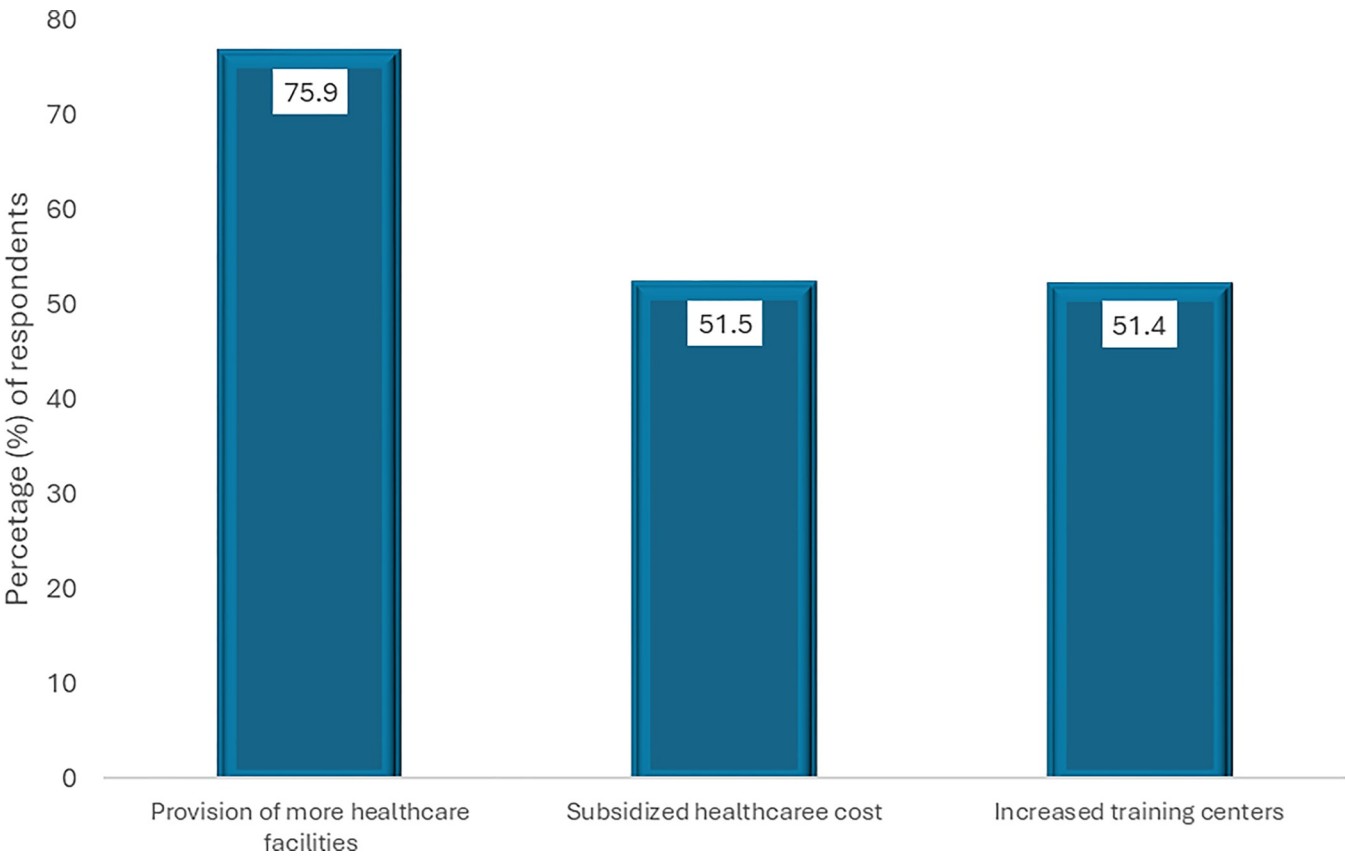

**Fig 2. Perceived areas of significant improvement over the years (n = 584 respondents).**

## Factors impacting the rating and attitude towards Nigerian healthcare

A multiple linear regression analysis was conducted to determine predictors of healthcare workers' ratings of the Nigerian healthcare system. Table 6 reveals that age and years of experience do not significantly predict healthcare workers' ratings, however, income ($p < 0.001$) and level of education ($p < 0.001$) emerged as significant positive predictors (unstandardized coefficient B = 2.434), thereby indicating that higher income and education levels correlate with more favourable ratings of the Nigerian healthcare system.

Table 7 represents a chi-square analysis of the relationship between nominal sociodemographic attributes and the average rating of healthcare financing, satisfaction with Nigerian government healthcare financing, and the motivation to continue working in healthcare. State of practice is significantly associated with all three rating categories ($p < 0.001$ for all), suggesting that the State where the healthcare workers are employed substantially impacts their views of healthcare financing and their motivation to continue working.

The professional designations have a significant association with satisfaction with healthcare financing ($p < 0.001$) and motivation to continue working ($p = 0.005$). This indicates that

**Table 5. Respondents' attitude towards continuation in the healthcare jobs and professions (n = 584).**

|  | Very Poor | Poor | Fair | Good | Excellent |
|---|---|---|---|---|---|
| Rate your motivation to continue working in a government healthcare facility. | 55 (9.4%) | 292 (50.0% | 180 (30.8%) | 44 (7.5%) | 13 (2.2%) |
| Rate your willingness to encourage others to join the healthcare professions in Nigeria. | 70 (12.0%) | 206 (35.3%) | 228 (39.0%) | 51 (8.7%) | 29 (5.0%) |

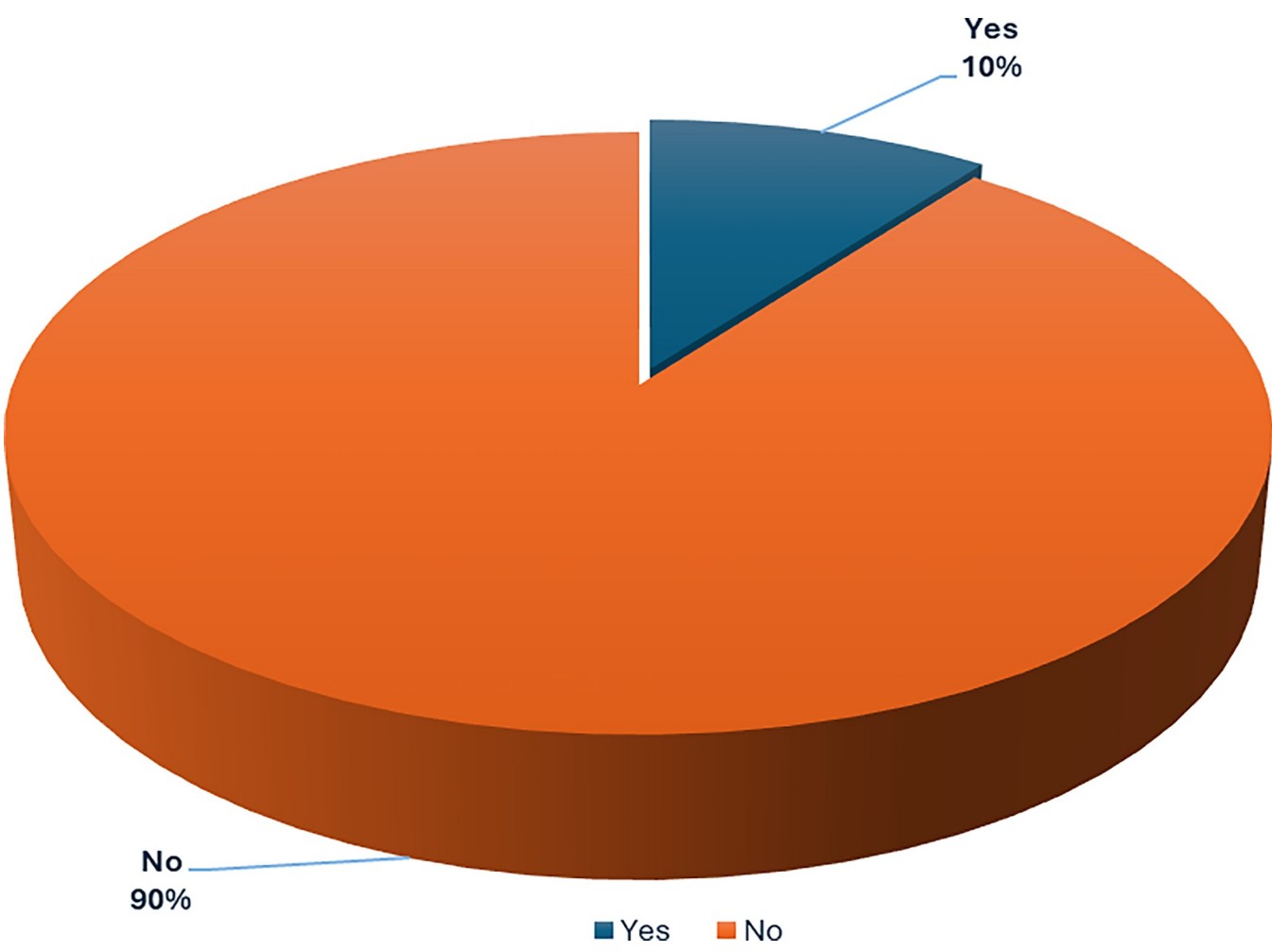

**Fig 3. Respondents satisfaction with the Nigerian healthcare system (n = 584).**

the respondent's specific type of healthcare profession influences their opinions and intentions about the healthcare system. Also, gender shares a significant relationship with satisfaction with healthcare financing (p <0.001), and employment type shows a weak association with motivation to continue working (p = 0.059) but no significant association with the other two rating categories.

Table 8 shows that most clinical healthcare professionals rated the healthcare financing system as either *poor* (61.3%) or *worse* (8.8%), indicating widespread dissatisfaction. While 22.9% rated it as *fair*, fewer (5.4%) respondents rated it as *good* or e*xcellent*. Physicians and nurses/midwives are more likely to rate the system as *poor* or *worse* compared to other professionals. On the other hand, physiotherapists and speech therapists had more favourable ratings of *fair* or *excellent*. The chi-square test reveals a significant association between clinical designation and ratings (p-value = 0.006), indicating that the differences in ratings across professional groups are not due to chance.

Table 9 presents a cross-tabulation analysis examining the relationship between the State of employment of all categories of healthcare workers and their ratings of the healthcare financing system. Many healthcare workers rated the healthcare finance and system as either *poor* (59.8%) or *worse* (8.2%). Also, 24.7% rated it as *fair*, while fewer healthcare workers rated it as

**Table 6. Regression analysis of the sociodemographic (ordinal) predictors of healthcare workers' rating of the Nigerian healthcare.**

| ANOVAª | | | | | |
|---|---|---|---|---|---|
| | **Sum of Squares** | **df** | **Mean Square** | **F** | **P-Value** |
| **Regression** | 28.992 | 4 | 7.248 | 13.588 | $< .001^b$ |
| **Residual** | 308.842 | 579 | 0.533 | | |
| **Total** | 337.834 | 583 | | | |

| Coefficients | | | | | |
|---|---|---|---|---|---|
| | **Unstandardized Coefficients** | | **Standardized Coefficients** | **T** | **P—Value.** |
| | **B** | **Std. Error** | **Beta** | | |
| **(Constant)** | 2.434 | 0.119 | | 20.410 | <0.001 |
| **Age Group** | -0.047 | 0.038 | -0.060 | -1.244 | 0.214 |
| **Education** | -0.135 | 0.029 | -0.200 | -4.652 | <0.001 |
| **Years of Experience** | 0.004 | 0.035 | 0.005 | 0.111 | 0.911 |
| **Income** | 0.128 | 0.025 | 0.213 | 5.103 | <0.001 |

a. Dependent Variable: Average rating of the healthcare financing.

b. Predictors: Income, Age Group, Education, Years of Experience.

*Significant at 95% confidence interval.

*good* (6.2%) or *excellent* (1.2%). The chi-square test reveals a significant association between the State of employment and ratings (p-value < 0.001), indicating that the differences in ratings across States are also not due to chance.

Table 10 shows that there is no strong evidence to suggest a significant relationship between non-clinical job roles and their ratings of the healthcare system's financing (p = 0.051).

Table 11 shows a statistically significant relationship between the average rating for healthcare financing and the motivation of healthcare workers to continue working in government health facilities (p < 0.001) with a positive coefficient of 0.670, suggesting that as the rating of healthcare financing improves, so does the motivation.

## Discussion

This study aimed to assess healthcare workers' perceptions of healthcare financing and systems quality in Nigeria over the immediate past period. It provides an important assessment of healthcare workers' perceptions of the Nigerian healthcare system, highlighting critical challenges and areas for improvement as valuable insights for policymakers, healthcare administrators, and stakeholders seeking to enhance the quality and accessibility of healthcare services in Nigeria. The sample was diverse, inclusive of various healthcare occupations within the clinical

**Table 7. Chi-square analysis of the sociodemographic (nominal) predictors of healthcare workers' rating of the Nigerian healthcare.**

| Chi-Square Tests (n = 584) | | | | | | | | | |
|---|---|---|---|---|---|---|---|---|---|
| **Factors** | **Average rating of the healthcare financing** | | | **Satisfaction with Nigerian Government Healthcare Financing** | | | **How Motivated are you to continue working in healthcare** | | |
| | **Value** | **df** | **P value** | **Value** | **df** | **P value** | **Value** | **df** | **P value** |
| Gender | 2.966 | 4 | 0.563 | 1053460.540 | 4 | < 0.001* | 2.645 | 4 | 0. 619 |
| State of Practice | 212.577 | 32 | < 0.001* | 1216426.074 | 18 | < 0.001* | 185.395 | 32 | < 0.001* |
| Employment type | 8.142 | 4 | 0. 086 | 1054161.883 | 4 | < 0.001* | 9.077 | 4 | 0. 059 |
| Professional designation | 7.740 | 4 | 0. 102 | 1049118.513 | 4 | < 0.001* | 14.944 | 4 | 0.005* |

*Significant at 0.05 significance level.

**Table 8. Relationship between the rating of the Nigerian healthcare system and the clinical designation of the clinical health (n = 445).**

| | | Average Ratings of the Healthcare Financing | | | | | Total n (%) | X² df P-Value |
|---|---|---|---|---|---|---|---|---|
| | | Worse n (%) | Poor n (%) | Fair n (%) | Good n (%) | Excellent n (%) | | |
| Clinical Designation | Emergency Medical Service | 0 (0) | 3 (42.9) | 4 (57.1) | 0 (0) | 0 (0) | 7 (100) | 61.079 36 0.006 |
| | Healthcare Assistant | 3 (5.4) | 37 (66.1) | 10 (17.9) | 6 (10.7) | 0 (0) | 56 (100) | |
| | Imaging (Radiographer, Sonographer etc.) | 1 (8.3) | 6 (50.0) | 4 (33.3) | 1 (8.3) | 0 (0) | 12 (100) | |
| | Laboratory Scientist | 3 (9.2) | 21 (63.6) | 7 (21.2) | 2 (6.1) | 0 (0) | 33 (100) | |
| | Nurse and/or Midwife | 13 (7.1) | 115 (62.8) | 49 (26.8) | 4 (2.2) | 2 (1.1) | 183 (100) | |
| | Pharmacist | 3 (7.3) | 27 (65.9) | 7 (17.1) | 3 (7.3) | 1 (2.4) | 41 (100) | |
| | Physician | 16 (21.6) | 41 (55.9) | 9 (12.2) | 4 (5.4) | 4 (5.4) | 74 (100) | |
| | Physiotherapist | 0 (0) | 15 (68.2) | 6 (27.3) | 1 (4.5) | 0 (0) | 22 (100) | |
| | Radiographer | 0 (0) | 8 (61.2) | 3 (23.1) | 2 (15.4) | 0 (0) | 13 (100) | |
| | Speech therapist | 0 (0) | 0 (0) | 3 (75.0) | 1 (25.0) | 0 (0) | 4 (100) | |
| Total | | 39 (8.8) | 273 (61.3) | 102 (22.9) | 24 (5.4) | 7 (1.6) | 445 (100) | |

*Significant at 0.05 significance level.

(76.2%) and non-clinical (24.8%) domains. Clinical staff included nurses/midwives, physicians, laboratory scientists, pharmacists and staff within the imaging units. The nonclinical staff included administration and management, cleaners (housekeeping), data management, front desk, records, driver or security, porter, kitchen, or maintenance, morgue attendant, social worker, and occupational therapists (S1 Table). Participants also comprised both genders, various age groups, and experience levels across the nine States.

## Healthcare workers' perception of the Nigerian healthcare finance system

The findings provide a clear picture of the prevailing sentiment among healthcare workers regarding healthcare financing in Nigeria. Most respondents (73.5%) expressed profound

**Table 9. Relationship between the State (Location) of employment and the rating of the Nigerian healthcare system (n = 584).**

| | | Average Ratings of the Healthcare Financing | | | | | Total n (%) | X², df P-Value |
|---|---|---|---|---|---|---|---|---|
| | | Worse n (%) | Poor n (%) | Fair n (%) | Good n (%) | Excellent n (%) | | |
| State | Delta | 10 (20.4) | 28 (57.1) | 11 (22.4) | 0 (0.0) | 0 (0.0) | 49 (100) | 212.577 32 <0.001* |
| | Enugu | 1 (2.9) | 31 (88.6) | 0 (0.0) | 2 (5.7) | 1 (2.9) | 35 (100) | |
| | Kaduna | 1 (1.5) | 27 (39.7) | 26 (38.2) | 13 (19.1) | 1 (1.5) | 68 (100) | |
| | Kano | 3 (3.2) | 46 (49.7) | 26 (28.0) | 16 (17.2) | 2 (2.2) | 93 (100) | |
| | Kwara | 22 (38.6) | 30 (52.6) | 4 (7.0) | 1 (1.8) | 0 (0.0) | 57 (100) | |
| | Lagos | 1 (1.0) | 59 (56.7) | 44 (42.3) | 0 (0.0) | 0 (0.0) | 104 (100) | |
| | Oyo | 3 (4.6) | 44 (67.7) | 14 (21.5) | 1 (1.5) | 3 (2.2) | 65 (100) | |
| | Plateau | 3 (9.7) | 18 (58.1) | 9 (29.0) | 1 (3.2) | 0 (0.0) | 31 (100) | |
| | Rivers | 4 (4.9) | 66 (80.5) | 10 (12.2%) | 2 (2.4) | 0 (0.0) | 82 (100) | |
| Total | | 48 (8.2) | 349 (59.8) | 144 (24.7) | 36 (6.2) | 7 (1.2) | 584 (100) | |

*Significant at 0.05 significance level.

**Table 10. Relationship between the non-clinical designation of employment and the rating of the Nigerian healthcare system (n = 584).**

| | | Average Ratings of the Healhtcare Financing | | | | | Total n (%) | $X^2$, df P-Value |
|---|---|---|---|---|---|---|---|---|
| | | Worse n (%) | Poor n (%) | Fair n (%) | Good n (%) | Excellent n (%) | | |
| Professional Designation | Administration or Management | 4 (15.4) | 11 (42.3) | 7 (26.9) | 4 (15.4) | 0 (0.0) | 26 (100) | 32.626 21 0.051 |
| | Cleaners/Housekeeping | 0 (0.0) | 9 (34.6) | 11 (42.3) | 6 (23.1) | 0 (0.0) | 26 (100) | |
| | Data management/Record, Front Desk | 2 (8.0) | 14 (56.0) | 8 (32.0) | 1 (4.0) | 0 (0.0) | 25 (100) | |
| | Driver or Security | 0 (0.0) | 16 (84.2) | 3 (15.8) | 0 (0.0) | 0 (0.0) | 19 (100) | |
| | Morgue Attendant | 1 (10.0) | 5 (50.0) | 4 (40.0) | 0 (0.0) | 0 (0.0) | 10 (100) | |
| | Occupational therapist | 0 (0.0) | 2 (50.0) | 1 (25.0) | 1 (25.0) | 0 (0.0) | 4 (100) | |
| | Porter, Kitchen staff, or Maintenance | 2 (10.5) | 13 (68.4) | 4 (21.1) | 0 (0.0) | 0 (0.0) | 19 (100) | |
| | Social Worker | 0 (0.0) | 6 (60.0) | 4 (40.0) | 0 (0.0) | 0 (0.0) | 10 (100) | |
| Total | | 9 (0.0) | 76 (54.7) | 42 (30.2) | 12 (8.6) | 0 (0.0) | 139 (100) | |

At 0.05 Significance level.

dissatisfaction with the current situation, reflected as a significant skew towards the poorer end of the rating scale (Table 4). This finding aligns with previous studies that consistently highlighted inadequate funding as a major constraint on the Nigerian healthcare system with suggestions that the poor ratings, especially of workforce skills and medical technology, are indicative of long-term issues such as insufficient training, research, and investment, implying a gradual decline or stagnation in quality amidst static budgets and increasing population with health demands [40–43].

## Prominent challenges in Nigerian healthcare systems

The healthcare workers highlighted multiple challenges in the Nigerian healthcare system, these were poor funding, poor services, poor staffing, lack of proper tools and facilities, lack of consumables and materials, fund misappropriation, and sub-standard pharmaceutical products.

**Table 11. Relationship between healthcare workers' rating of the Nigerian healthcare system and their motivation to continue working in government health facilities.**

| ANOVA[a] | | | | | |
|---|---|---|---|---|---|
| | Sum of Squares | df | Mean Square | F | p-Value |
| Regression | 151.783 | 1 | 151.783 | 330.262 | <0.001[b] |
| Residual | 267.477 | 582 | 0.460 | | |
| Total | 419.260 | 583 | | | |

| Coefficients | | | | | |
|---|---|---|---|---|---|
| | Unstandardized Coefficients | | Standardized Coefficients | t | p-Value |
| | B | Std. Error | Beta | | |
| (Constant) | 0.874 | 0.090 | | 9.692 | <0.001 |
| Average Ratings of the Healthcare Financing | 0.670 | 0.037 | 0.602 | 18.173 | <0.001* |

a. Dependent Variable: How motivated are you to continue working in government health facilities?.

b. Predictors: (Constant), Average Ratings of the Healthcare Financing.

*Significant at 0.05 (95% confidence interval).

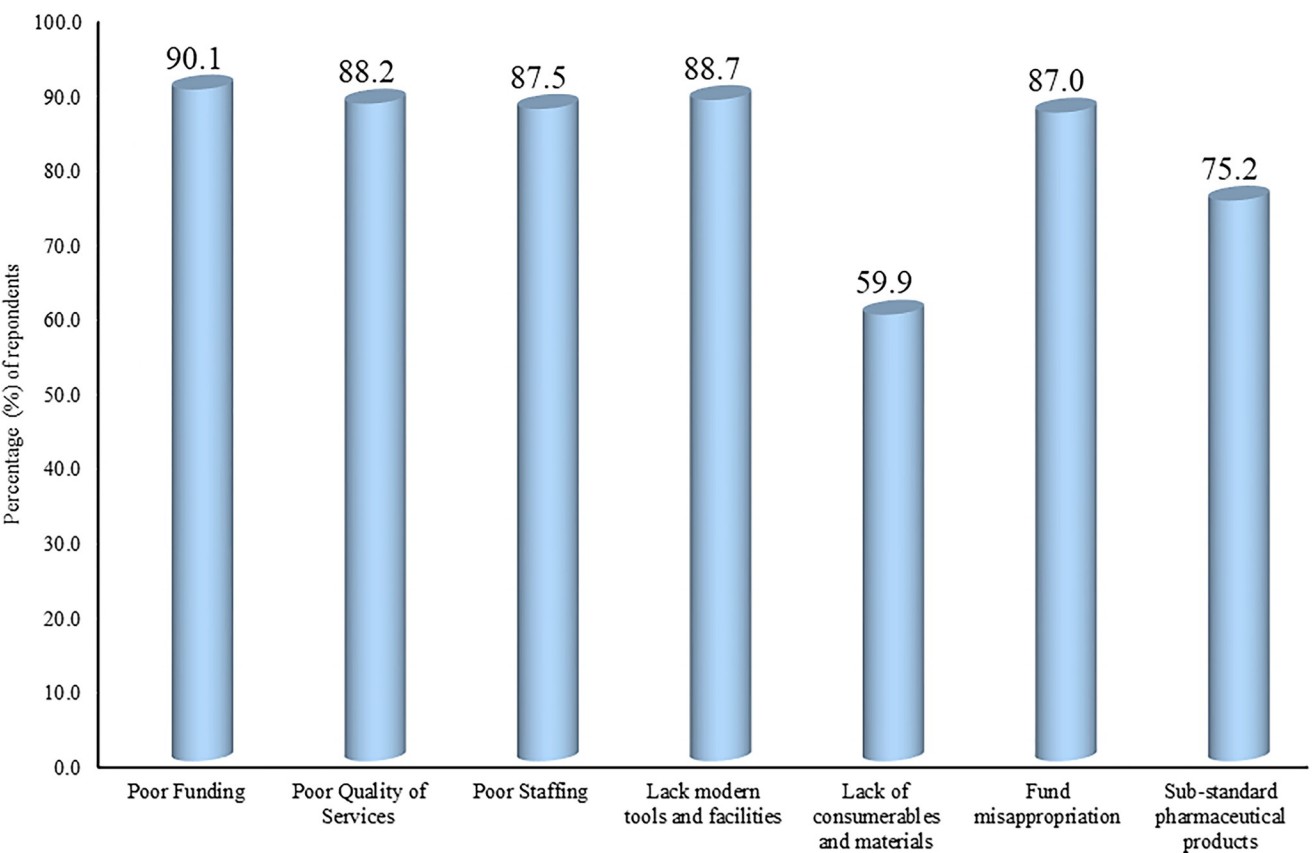

**Fig 4. Perceived major problem areas in the Nigerian healthcare system (n = 584 respondents).**

**Poor funding.**   Poor funding in Nigerian healthcare was identified as a major challenge in the healthcare system. This challenge is complex with multiple interrelated factors tied to inadequate government budgetary allocation, inefficient resource management, the predominance of out-of-pocket payments, weak revenue generation, limited institutional capacity, economic challenges, and a lack of political will to intervene, which all contribute to chronic underfunding [44–48]. Nigeria's healthcare expenditure is significantly below the recommended 5% of GDP and this low funding has resulted in a chronic shortage of resources for essential services.

Also, a significant portion of healthcare costs in Nigeria is borne by patients out-of-pocket, which can lead to financial hardship and limit access to quality care among those with limited resources [12–14]. This reliance on out-of-pocket payments can also hinder the development of a sustainable healthcare financing system as it plunges users into more financial crises. The challenges are further compounded by the unstable Nigerian economy, the weak revenue generation (poorly structured taxation and other internally generated revenue measures), limited institutional capacity (weak governance, inefficient management practices, and a shortage of qualified healthcare professionals), and lack of political commitment to increase funding to healthcare [45–48].

**Poor quality of services.**   While poor funding is a pervasive challenge in the Nigerian healthcare system, poor quality of care is one of several other critical issues highlighted by healthcare workers that must also be addressed to improve the system (Fig 4). Dissatisfactory quality of healthcare services may be attributable to factors such as insufficient staffing, lack of essential equipment, lack of training, and inefficient process management [49].

**Poor staffing.**    A third concern was poor staffing, which was seen by a significant number of healthcare workers as a major challenge. This is defined by attributes like shortages of qualified professionals, inadequate training, and high attrition rates [49]. Without sufficient and well-trained healthcare professionals, it is difficult to provide quality care and maintain the health facilities to meet the population's needs [50]. It is, therefore, important to address this issue by investing in human resources development and creating a more supportive work environment as partly enshrined in the Nigerian Federal Ministry of Health's 2023 National Health Policy on Health Workforce Migration [44,51,52].

**Lack of proper/modern tools and facilities.**    Healthcare workers also reported a lack of proper tools and facilities, such as diagnostic equipment, medications, and infrastructure as a major challenge in healthcare delivery. This has hindered the effective delivery of healthcare services and compromised patient safety especially the lack of access to essential diagnostic equipment/tools as seen with the prevalent misdiagnosis of typhoid fever among patients in Murtala Muhammad Specialist Hospital Kano due to the use of outdated diagnostic methods [44,53,54].

**Lack of consumables and essential materials.**    Healthcare providers expressed concerns over the lack of basic consumables. The lack of consumable and essential care materials, such as syringes, bandages, and gloves, can reduce care effectiveness, compromise infection control practices, and increase the risk of healthcare-associated infections. These challenges can have a devastating impact on patient health and undermine the effectiveness of healthcare delivery [44].

**Sub-standard pharmaceutical products.**    The availability of substandard pharmaceuticals in Nigerian healthcare facilities is a significant challenge that can have severe consequences for patient health [55]. Substandard pharmaceuticals, which may be counterfeit, adulterated, or mislabeled, can lead to a range of adverse health outcomes. For example, counterfeit drugs may contain incorrect or insufficient active ingredients, leading to treatment failure or ineffective therapy. Adulterated drugs may be contaminated with harmful substances, such as heavy metals or toxins, which can cause serious health problems, while mislabeled drugs may have incorrect dosage instructions or be packaged in misleading containers, increasing the risk of medication errors and adverse drug reactions [55].

**Poor facility maintenance.**    Healthcare workers lamented that where the facilities exist, they often lay to waste and become outdated due to poor maintenance practices. This further complicates the issues and compromises the quality of healthcare delivery just as identified by Oladejo et al [50].

**Fund misappropriations.**    Concerns about fund misappropriations were raised by healthcare workers, highlighting the need for improved financial management practices and increased transparency and accountability in the healthcare sector. Corruption, mismanagement, and a lack of transparency contribute to the inefficient use of available resources and can result in funds being diverted from essential services, thereby creating shortages of equipment, medications, and manpower. It can also erode public trust in the healthcare system and undermine efforts to improve healthcare delivery [56].

There is a pressing need for increased funding and optimal management processes to tackle pervasive issues and modernize healthcare in Nigeria as respondents perceive extensive shortfalls in healthcare infrastructure, technology, tools, and personnel, which mostly hinge on resource scarcity following episodes of chronic underinvestment that is further compounded by the fast-growing population, increasing public health burden, and struggling national and regional economy [8,20,57–59]. Although the Nigeria government has made attempts toward addressing the challenges in the healthcare systems, the core challenges of fund management and quality assurance continue to linger in the entire system [12,60,61]. According to

Onwujekwe et al. [20] the existing healthcare financing mechanisms in Nigeria fail to deliver expected outcomes due to poor subscription to evidence-based resources and approach. They, therefore, recommended revision of the legal and regulatory frameworks for better health financing, since multiple evidence have identified funding and workforce shortages as barriers to care quality [62–64].

## Perceived areas with significant improvements

Regardless of the challenges identified, our findings offer some positive perspective on the Nigerian healthcare system, highlighting several key improvements. Most healthcare workers (75.9%) reported an increase in healthcare facilities, indicating progress in expanding access to care. This expansion is likely due to government initiatives to construct new facilities and renovate existing ones and investments from private sector entities. In addition, a substantial but lower proportion of respondents (51.5%) commended government effort towards subsidizing healthcare costs. These subsidies, which mainly include the National Health Insurance (NHIS) and other special healthcare programs can help reduce out-of-pocket expenses for patients, making healthcare more affordable and accessible which is particularly important for low-income individuals and families who may struggle to afford necessary medical care [65,66]. It is also important to note that while this is seen as a major stride towards better healthcare, the insurance enrolment rate is still very low across the country [67].

Furthermore, the increase in training centres for healthcare professionals is identified by 51.4% of healthcare workers as a positive development seen within the immediate past period. This suggests that the government may be investing more in human resources development and recognizing the importance of having a well-trained healthcare workforce. By increasing the number of training centres and continuing education opportunities, more individuals can acquire the necessary skills and qualifications to provide quality care as typified in the notable example of upgrade and expansion of nursing and medical training across the country, through several initiatives and partnerships like the Global Health Workforce Programme (GHWP) [52,68]. This is important because addressing the challenges of healthcare quality requires a comprehensive and sustained effort from both the government and stakeholders to ensure that the healthcare system can meet the needs of the population.

## HCWs' satisfaction and motivation with the Nigerian healthcare system

**HCWs' satisfaction with the Nigerian healthcare system.**   We found a clear assessment of healthcare workers' perceptions of the Nigerian healthcare system. A substantial majority of respondents (90%) indicated dissatisfaction with the current state of the system, while only a small minority (10%) expressed satisfaction (Fig 3). The low satisfaction levels reported by healthcare workers underscore the urgent need for comprehensive reforms to improve the quality and accessibility of healthcare services in Nigeria as addressing these challenges requires a multifaceted approach that should include increasing healthcare funding, investing in infrastructure, strengthening human resources, and addressing regional disparities.

**HCWs' motivations and attitude towards their jobs.**   Our findings further provide insights into the motivation and recruitment efforts (encouraging others to join the profession) among healthcare workers in Nigeria. The data reveals a concerning trend as the significant majority (59.4%) of healthcare workers have low motivation to continue working in government healthcare facilities, while there is a mixed response to encouraging others to join the profession as 47.3% of the workers are not motivated to do so (Table 5). The high proportion of demotivated healthcare workers is a significant cause for concern which suggests that many healthcare professionals are not satisfied with their current jobs and may be considering

leaving the profession and field altogether. These findings are similar to findings by Ebuehi and Campbell [53] who surveyed 200 healthcare workers in Ogun state and found that poor financing and other factors were significant motivations for staff intentions and retention. Ebuehi and Campbell [53] identified a range of significant challenges within the Nigerian healthcare sector, including inadequate funding, poor infrastructure, limited access to essential resources, and career growth as seen as key to staff motivation similar to Bhatnagar [69] who found that "both intrinsic (self-efficacy, religion, choice of profession) and extrinsic (good working environment including supportive supervision, monetary incentives, recognition, organizational justice) factors" influenced motivation among healthcare professionals in Nigeria and India [69].

The impacts of poor healthcare system financing and management are numerous. A broad example is *poor working conditions* in which healthcare workers are faced with challenging conditions, such as long hours, heavy workloads, exposure to hazardous materials, and lack of basic amenities like electricity and water, leading to increased malpractice [70], burnout, stress, and job dissatisfaction as seen by Weldegebriel et al [71] in a motivation study conducted among eight public hospitals in Ethiopia in 2013. Another typical example is *inadequate remuneration* which hinges on the fact that low salaries and limited opportunities for career advancement can make it difficult for healthcare workers to afford basic living expenses and achieve professional goals. This can lead to a lack of motivation and a desire to seek employment elsewhere since the salaries offered to healthcare workers in Nigeria is not commensurate with the demanding nature of their work and the critical role they play in society [72,73]. In addition, *limited resources and infrastructure* due to insufficient funding can create a challenging and frustrating work environment that affects healthcare workers' morale and job satisfaction. As seen in our study, healthcare facilities are reported to often lack essential infrastructure, modern equipment, efficient supplies, and quality medications, thereby hindering the effective delivery of care promoting frustration among healthcare workers and potentially compromising patient safety [74,75].

### Factors influencing healthcare workers' perception of healthcare finance and quality

**Sociodemographic predictors.**  The regression analysis presented in Table 6 highlights the significant influence of income and education levels on healthcare workers' ratings of the Nigerian healthcare system. Higher education level and income are associated with more favourable perceptions, suggesting that socioeconomic factors play a crucial role in shaping healthcare workers' views [76,77]. The chi-square analysis (Table 7) reveals that the state in which healthcare workers are employed has a substantial impact on their views of healthcare financing, satisfaction with Nigerian government healthcare financing, and motivation to continue working. This suggests that regional disparities in healthcare quality, funding, and working conditions influence healthcare workers' perceptions and intentions. While it is important to create a healthcare climate of patient-centeredness, the policymakers must bear in mind the disparities in resources, living costs, and other factors that influence the actual implementation of the healthcare process in various states and regions [44,75,78].

**Job and employment attributes.**  *State (Location) of employment*. A cross-tabulation analysis examining the relationship between the state of employment of healthcare professionals and their ratings of the Nigerian healthcare financing system. The results indicate that healthcare professionals are dissatisfied with the current system, with over 50% rating it as *poor* or *worse*. However, there are notable variations across different states, suggesting regional disparities in the quality of healthcare financing. The chi-square test reveals a significant association

between the state of employment and ratings. While the overall trend is negative, there are notable variations across different states: healthcare professionals in Delta and Kwara states are more likely to rate the system as *worse* or *poor*, while those in Kaduna and Kano states have a more balanced distribution of ratings. This shows that healthcare systems in the southern part of Nigeria may be in poorer states than the north, thereby amplifying the need to re-evaluate the funding mechanisms and fund management for healthcare improvements in various regions.

*Professional designations.* Professional designation and employment type exhibit weak associations with the healthcare workers' perceptions. Whether one is clinical, or nonclinical healthcare role is significantly associated with satisfaction with Nigerian government healthcare financing and motivation to continue working, implying that the category of healthcare job can influence views and intentions. Data further suggests that more clinical staff rate the healthcare system more poorly than the non-clinical staff. This is likely because clinical staff are likely to be more directly affected by the prevailing conditions of the healthcare systems.

*Clinical designation.* While there was no significant relationship between non-clinical roles and their ratings of the healthcare system, a cross-tabulation analysis examining the relationship between clinical designation and ratings of the Nigerian healthcare financing system showed that many healthcare professionals are dissatisfied with the current system, with over 60% rating it as *poor*. However, there are notable variations across different professions, with physicians and nurses expressing the highest levels of dissatisfaction while physiotherapists and speech therapists report more favourable ratings. However, 1.1%, 2.4%, and 5.4% of Nurses and/or Midwives, Pharmacists, and Physicians rated the healthcare systems as excellent, respectively. This indicates that even among the key professions in frequent direct contact with the patients the Physicians reported more satisfaction with the healthcare system five times as much as Nurses and twice more than the Pharmacists. Overall, the chi-square test reveals a significant association between clinical designation and ratings, suggesting that the differences in perceptions are not due to chance. These findings highlight the need for policy reforms to address the systemic issues contributing to dissatisfaction among healthcare professionals and improve the overall quality of healthcare in Nigeria. Further research is necessary to explore the factors underlying these variations in perceptions and inform targeted interventions to enhance the healthcare system.

*Employment type.* Also, employment type (full-time or part-time) shows a weak association with motivation to continue working but no significant association with the other rating categories. These imply that one's type of job as well as job security are major factors impacting the attitudes and motivation of healthcare workers towards the healthcare system as reported by Jibril et al [79], Ebuehi and Campbell [53], Okeke et al [72], and Weldegebriel et al [71]. Our study, however, adds insights into the professional roles as key determinants of healthcare workers perception and satisfaction with the healthcare finance and system in Nigeria.

*Healthcare financing on healthcare worker motivation.* Our data demonstrates a strong positive relationship between the average rating of healthcare financing and the motivation of healthcare workers to continue working in government health facilities (Table 11). This suggests that improving healthcare financing can significantly enhance healthcare workers' job satisfaction and improve staff retention, thereby elevating the quality of care delivery [80,81].

## Public health implications and policy recommendations

To improve healthcare workers' satisfaction and retention and promote optimum care quality, it is crucial to address the socioeconomic disparities that influence their perceptions. This should involve the following.

- There is need for policy reforms to address the systemic issues contributing to dissatisfaction among healthcare professionals and improve the overall quality of healthcare in Nigeria.

- Increasing funding for healthcare, improving working conditions, and providing opportunities for career advancement, particularly in underserved reg

- ions. In addition, investing in human resources development and ensuring equitable access to education and training can help create a more satisfied and motivated healthcare workforce.

- Policymakers are urged to make targeted investments in essential healthcare delivery areas currently facing deficits [82]. It's crucial for policies to support the adoption of advanced technologies, innovative service models, and a rejuvenated workforce [83].

- As the population and disease patterns evolve, budgets should proactively support new access and quality improvement initiatives [84].

- Decision-making should be data-driven and include professional input, ensuring a strategic and proactive approach towards budgeting. Health system decisions must also reflect the specific needs of Nigerians who rely on public healthcare services, with health campaigns and investments tailored to Nigeria's diverse population [85,86].

- A 10-year strategic health infrastructure plan is recommended to direct this funding towards addressing critical shortages, from medical equipment to staff retention, including the need for a significant increase in public health funding to meet the Abuja declaration's 15% of the budget for health goals.

- Encouraging public-private partnerships to decentralize quality care through new local primary facilities could quickly improve infrastructure and service capacity. These health financing reforms are vital for enhancing the quality, reach, and efficiency of Nigerian healthcare to yield sustainable and efficient universal health coverage.

- The funding of healthcare should also be further supported by the federal and state governments using need assessments, *transparent fund tracking*, and evidence-based methodologies of purchasing such as the Strategic Purchasing African Resources Center (SPARC) framework can help optimize the allocation and use of healthcare funds [60,70].

## Limitations

The study, while shedding light on healthcare workers' perspectives on Nigerian healthcare challenges, acknowledges its limitations, such as exclusive focus on healthcare staff views, and excluding broader patient/public input. It also did not look at differences within the clinical healthcare workers. We have engaged in and encouraged specific research that covers the patients and public focusing on specific budgeting, healthcare utilization, and suggestions for system improvements among the general Nigerian populace. Finally, the research is subject to various biases associated with cross-sectional surveys such as recall bias, selection bias, and minimal situational interferences were minimized by survey question optimization as well as response randomization and anonymization.

## Conclusion

This research provides a comprehensive assessment of healthcare workers' perceptions of healthcare financing and quality in Nigeria. The findings highlight significant challenges

within the system, including poor funding, inadequate infrastructure, insufficient staffing, and limited access to essential resources. These challenges contribute to low job satisfaction, demotivation, and a desire to leave the profession among healthcare workers.

The perception of the healthcare workers in Nigeria about the healthcare systems vary by professional designation (clinical vs nonclinical) clinical designation (profession), and employment type (full-time vs part-time) reflecting the need to mitigate the systemic issues that promote the disparities.

To address these issues, the study recommends increasing healthcare funding, investing in infrastructure, strengthening human resources development, and addressing regional disparities. By implementing these reforms, Nigeria can improve the quality and accessibility of healthcare services, enhance healthcare workers' satisfaction and retention, and ultimately improve the health and well-being of its citizens.

## Supporting information

**S1 Checklist. Inclusivity in global research.** S1 Checklist is a PLOS' policy on inclusivity in global research which focuses on ensuring that research conducted outside of a researcher's home country is reported transparently and ethically.
(DOCX)

**S1 Table. Professional designations of respondents.** S1 Table provides a breakdown of the professional designations of the 584 Nigerian health workers who responded to this survey. Majority of the respondents are involved in clinical roles, with nurses and midwives being the most represented profession. However, non-clinical roles such as administration, cleaning, and data management are also significantly represented.
(DOCX)

## Acknowledgments

We extend our deepest gratitude to the dedicated team of individuals who contributed their time and expertise to the data collection phase of our research. Special thanks to Abosede Peace Adebayo, Deborah Oyinlola Salawu, Busiroh Mobolape, Ibraheem, Ubiebo Ataisi Ekenekot, Precious Ebinehita Imoyera, Mudiaga Sidney Edafiejire, Joy Chioma Obialor, Gloria Oluwakorede Alao, Blessing Onyinye Obialor, and Ajao Adewale Gbolabo. Your commitment and meticulous efforts were instrumental in the success of our study. We are immensely appreciative of your contributions.

## Author Contributions

**Conceptualization:** Blessing Osagumwendia Josiah, Brontie Albertha Duncan, France Ncube, Timothy Wale Olaosebikan, Marios Kantaris.

**Data curation:** Blessing Osagumwendia Josiah, Emmanuel Chukwunwike Enebeli, Lordsfavour Uzoma Anukam, Blessing Chiamaka Nganwuchu.

**Formal analysis:** Blessing Osagumwendia Josiah, Emmanuel Chukwunwike Enebeli, Brontie Albertha Duncan, Lordsfavour Uzoma Anukam, Ndidi Louis Otoboyor.

**Funding acquisition:** Blessing Osagumwendia Josiah, Emmanuel Chukwunwike Enebeli, Brontie Albertha Duncan, Lordsfavour Uzoma Anukam, Chinelo Cleopatra Josiah, Eric Kelechi Alimele, Oghosa Gabriel Josiah, Jemima Ufuoma Mukoro.

**Investigation:** Blessing Osagumwendia Josiah, Emmanuel Chukwunwike Enebeli, Brontie Albertha Duncan, Lordsfavour Uzoma Anukam, Oluwadamilare Akingbade, Chinelo Cleopatra Josiah, Jemima Ufuoma Mukoro, Blessing Chiamaka Nganwuchu.

**Methodology:** Blessing Osagumwendia Josiah, Emmanuel Chukwunwike Enebeli, France Ncube, Fawole Israel Opeyemi, Marios Kantaris.

**Project administration:** Blessing Osagumwendia Josiah, Emmanuel Chukwunwike Enebeli.

**Resources:** Blessing Osagumwendia Josiah, Emmanuel Chukwunwike Enebeli, Eric Kelechi Alimele, Ndidi Louis Otoboyor, Oghosa Gabriel Josiah, Blessing Chiamaka Nganwuchu, Fawole Israel Opeyemi.

**Supervision:** Blessing Osagumwendia Josiah, Emmanuel Chukwunwike Enebeli, France Ncube, Timothy Wale Olaosebikan, Marios Kantaris.

**Validation:** Blessing Osagumwendia Josiah, Emmanuel Chukwunwike Enebeli, Brontie Albertha Duncan, Lordsfavour Uzoma Anukam, Oluwadamilare Akingbade, France Ncube, Fawole Israel Opeyemi, Timothy Wale Olaosebikan, Marios Kantaris.

**Visualization:** Blessing Osagumwendia Josiah, Emmanuel Chukwunwike Enebeli, Jemima Ufuoma Mukoro, Marios Kantaris.

**Writing – original draft:** Blessing Osagumwendia Josiah, Emmanuel Chukwunwike Enebeli, Brontie Albertha Duncan, Lordsfavour Uzoma Anukam, Oluwadamilare Akingbade, France Ncube, Chinelo Cleopatra Josiah, Eric Kelechi Alimele, Ndidi Louis Otoboyor, Oghosa Gabriel Josiah, Timothy Wale Olaosebikan, Marios Kantaris.

**Writing – review & editing:** Blessing Osagumwendia Josiah, Emmanuel Chukwunwike Enebeli, Brontie Albertha Duncan, Lordsfavour Uzoma Anukam, Oluwadamilare Akingbade, Chinelo Cleopatra Josiah, Eric Kelechi Alimele, Ndidi Louis Otoboyor, Oghosa Gabriel Josiah, Fawole Israel Opeyemi, Marios Kantaris.

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
