## [Decision Letter · Decision Letter 0]

30 Jul 2024

PGPH-D-24-01007

Perceptions of Healthcare Finance and Quality Among Nigerian Healthcare Workers

Dear Dr. Josiah,

Thank you for submitting your manuscript to PLOS Global Public Health. After careful consideration, we feel that it has merit but does not fully meet PLOS Global Public Health’s publication criteria as it currently stands. Therefore, we invite you to submit a revised version of the manuscript that addresses the points raised during the review process.

Please note that we have only been able to secure a single reviewer to assess your manuscript. We are issuing a decision on your manuscript at this point to prevent further delays in the evaluation of your manuscript. Please be aware that the editor who handles your revised manuscript might find it necessary to invite additional reviewers to assess this work once the revised manuscript is submitted. However, we will aim to proceed on the basis of this single review if possible. 

We look forward to receiving your revised manuscript.

Kind regards,

Avanti Dey, PHD

Staff Editor

Journal Requirements:

Additional Editor Comments (if provided):

Reviewers' comments:

Reviewer's Responses to Questions

**Comments to the Author**

1. Does this manuscript meet PLOS Global Public Health’s publication criteria? Is the manuscript technically sound, and do the data support the conclusions? The manuscript must describe methodologically and ethically rigorous research with conclusions that are appropriately drawn based on the data presented.

Reviewer #1: Yes

2. Has the statistical analysis been performed appropriately and rigorously?

Reviewer #1: I don't know

3. Have the authors made all data underlying the findings in their manuscript fully available (please refer to the Data Availability Statement at the start of the manuscript PDF file)?

Reviewer #1: Yes

4. Is the manuscript presented in an intelligible fashion and written in standard English?

Reviewer #1: Yes

5. Review Comments to the Author

Reviewer #1: Thank you for the opportunity to review this valuable manuscript. The study's objective of evaluating the quality of the health system in Nigeria is of great significance. It provides crucial insights that can greatly benefit policymakers in Nigeria and other countries with similar healthcare infrastructures. Here are some recommendations to enhance the transparency of the study and make it even more informative about improving the quality of the health system.

Methods:

Overall, I missed some key information about the data collection tool, the type of health providers included, the level of healthcare providers (primary, secondary, tertiary), the study area, the population, and the data analysis. See specific points below:

1. Did you consider heterogeneity in health worker density across states and urban/rural scenarios to calculate the sample size? For example, Kano has the biggest total population, but Lagos may have a higher density of health workers despite its smaller population. So, would this change the sample size estimate?

2. Can you provide more information about the study population and inclusion criteria, including clinical and non-clinical categories: doctors, nurses, and technicians? The same is true for study settings (urban/rural) and types of health facilities (primary, secondary, tertiary level; public, private; outpatient, inpatient care). Please clarify the inclusion criteria. Also, how did you select the health units and workers in each state?

3. It is unclear how the sample of 600 health workers was reached. I understand you tried to capture the difference between clinical and non-clinical staff, but how was these additional figures calculated?

4. Include more information about the data collection tool; for example, was it based on previous studies? Also, include information on how many health workers were included to test the questionnaire's reliability.

5. Can you please justify using the Pearson correlation and which correlation you are looking at? The same is true for regression analysis. Which regression did you use, which variables (exposure and explanatory), and why was this test applied? It seems that regression is more useful in finding associations, but the article states that it was applied to ‘exploring variations in perceptions across demographics’. I could not find these ‘variations’ in the results section.

Results:

1. Did you find any differences in satisfaction across states or even between clinical and non-clinical professionals? It would be an interesting finding to explore.

1. Elaborate more on the description of Table 3: include frequencies and percentages and the average percentage for each grade (E, G, F, P, VP). Also, Include frequencies in this Table.

2. Include labels in Figures 3 and 4.

2. This sentence should be in the methods section: ‘In the study’s Likert scale evaluation of the Nigerian healthcare system, ratings were assigned as follows: 5 for Excellent, 4 for Good, 3 for Fair, 2 for Poor, and 1 for Very Poor. The cumulative scores of participants were then categorized into ranges: 6-10 for Very Poor, 11-15 for Poor, 16- 20 for Fair, 21-25 for Good, and 26-30 for Excellent.’

3. There is no description of Figure 3.

4. Please clarify what Figure 2 represents. For example, what do you mean by satisfaction?

5. The description of the Pearson correlation seems to be misinterpreted. The correlation of 0.5 shows that there is a ‘moderate’ positive correlation between rating and satisfaction, not that satisfaction tends to increase alongside higher ratings.

6. Overall, I could not understand the regression analysis, mainly because of the lack of information in the methods section. Can you clarify the variables you applied in Table 6? For example, what is the reference category for each variable?

Discussion:

This section needs a little more work. At the moment, each paragraph often summarises the findings from this study and then references other studies elsewhere. Still, the link between the two isn’t always explicit, and the implications for the Nigerian health system aren’t always stated. See more details below.

1. The statement’ The potential skew towards nursing priorities due to their overrepresentation is noted’, indicates that it is important to include this variable in Table 2. Also, this statement pertains to the limitations section.

Perceived Quality of Healthcare:

3. Why would an increase in government healthcare subsidies translate to improvement in healthcare worker's perspective? What is this statement based on? How about the other improvements? Why have they not been reflected in improving the quality of the health system?

4. Insufficient training was reported as a long-term issue for poor ratings. However, increasing the number of training centres was one of the improvements reported (~51%), which contradicts this statement.

5. Patient perception is not the focus of this study, so how can you state that there is a discrepancy between patients and health workers' perceptions?

Most prominent challenges

6. Can you include more details of the previous studies when you mention them: setting, focus (specific health conditions or healthcare facilities)?

7. Table 3, not Figure 2, highlights the most critical challenges in Nigerian healthcare as identified by healthcare workers.

8. Brain drain was not reported in the results section, but the discussion says that 88.7% of respondents reported it as an issue. Please include it in the results.

Public Health Implications and Policy Recommendations

9. This section brings several recommendations, all very important and with an important impact on improving the health system's quality. However, more reflection on the feasibility of implementing all these policies/interventions in a lower-middle-income country would make this section more beneficial from the policy perspective.

Conclusions

The study's focus on the lack of appropriate funding overshadows the other challenges reported with similar frequency. I recommend reflecting on the overall challenges and how these findings can guide the changes the health workers want to see across the health system in Nigeria.

6. PLOS authors have the option to publish the peer review history of their article (what does this mean?). If published, this will include your full peer review and any attached files.

**Do you want your identity to be public for this peer review?** For information about this choice, including consent withdrawal, please see our Privacy Policy.

Reviewer #1: **Yes: **Noemia Siqueira

---

## [Decision Letter · Decision Letter 1]

8 Oct 2024

Perceptions of Healthcare Finance and System Quality Among Nigerian Healthcare Workers

PGPH-D-24-01007R1

Dear Mr Josiah,

We are pleased to inform you that your manuscript 'Perceptions of Healthcare Finance and System Quality Among Nigerian Healthcare Workers' has been provisionally accepted for publication in PLOS Global Public Health.

Best regards,

Yuvaraj Krishnamoorthy

Academic Editor

Reviewer Comments (if any, and for reference):

Reviewer's Responses to Questions

**Comments to the Author**

1. If the authors have adequately addressed your comments raised in a previous round of review and you feel that this manuscript is now acceptable for publication, you may indicate that here to bypass the “Comments to the Author” section, enter your conflict of interest statement in the “Confidential to Editor” section, and submit your "Accept" recommendation.

Reviewer #1: All comments have been addressed

2. Does this manuscript meet PLOS Global Public Health’s publication criteria? Is the manuscript technically sound, and do the data support the conclusions? The manuscript must describe methodologically and ethically rigorous research with conclusions that are appropriately drawn based on the data presented.

Reviewer #1: Yes

3. Has the statistical analysis been performed appropriately and rigorously?

Reviewer #1: Yes

4. Have the authors made all data underlying the findings in their manuscript fully available (please refer to the Data Availability Statement at the start of the manuscript PDF file)?

Reviewer #1: Yes

5. Is the manuscript presented in an intelligible fashion and written in standard English?

Reviewer #1: Yes

6. Review Comments to the Author

Reviewer #1: The authors addressed all comments and have made major improvement in the report of the data analysis, results and discussion sections.

7. PLOS authors have the option to publish the peer review history of their article (what does this mean?). If published, this will include your full peer review and any attached files.

**Do you want your identity to be public for this peer review?** For information about this choice, including consent withdrawal, please see our Privacy Policy.

Reviewer #1: **Yes: **Noemia Siqueira
